# Geographical distribution of Enterobacterales with a carbapenemase IMP-6 phenotype and its association with antimicrobial use: An analysis using comprehensive national surveillance data on antimicrobial resistance

Aki Hirabayashi[1]☉*, Koji Yahara[1]☉*, Toshiki Kajihara[1], Motoyuki Sugai[1], Keigo Shibayama[1,2]

**1** Antimicrobial Resistance Research Center, National Institute of Infectious Diseases, Tokyo, Japan,
**2** Department of Bacteriology II, National Institute of Infectious Diseases, Tokyo, Japan

☉ These authors contributed equally to this work.
* akihira@nih.go.jp (AH); k-yahara@nih.go.jp (KY)

**Data Availability Statement:** All data to replicate the study's findings are fully available at https://

## Abstract

Enterobacterales resistant to carbapenems, a class of last-resort antimicrobials, are ranked as an "urgent" and "critical" public health hazard by CDC and WHO. IMP-type carbapenemase-containing Enterobacterales are endemic in Japan, and $bla_{IMP-6}$ is one of the notable carbapenemase genes responsible for the resistance. The gene is plasmid-encoded and confers resistance to meropenem, but not to imipenem. Therefore, IMP-6-producing Enterobacterales isolates are occasionally overlooked in clinical laboratories and are referred to as 'stealth-type'. Since previous reports in Japan were confined only to some geographical regions, their distribution across prefectures and the factors affecting the distribution remain unclear. Here, we revealed the dynamics of the geographical distribution of Enterobacterales with IMP-6 phenotype associated with antimicrobial use in Japan. We utilized comprehensive national surveillance data of all routine bacteriological test results from more than 1,400 hospitals in 2015 and 2016 to enumerate *Escherichia coli* and *Klebsiella pneumoniae* isolates with the antimicrobial susceptibility pattern (phenotype) characteristic of IMP-6 (imipenem susceptible, meropenem resistant), and to tabulate the frequency of isolates with the phenotype for each prefecture. Isolates were detected in approximately half of all prefectures, and combined analysis with the national data of antimicrobial usage revealed a statistically significant association between the frequency and usage of not carbapenems but third-generation cephalosporins ($p = 0.006$, logistic mixed-effect regression) and a weaker association between the frequency and usage of fluoroquinolones ($p = 0.043$). The usage of third-generation cephalosporins and fluoroquinolones may select the strains with the IMP-6 phenotype, and contribute to their occasional spread. We expect the findings will promote antimicrobial stewardship to reduce the spread of the notable carbapenemase gene.

github.com/bioprojects/IMP-6_phenotype_
distribution.

**Funding:** This study was supported by Research
Program on Emerging and Re-emerging Infectious
Diseases from the Japan Agency for Medical
Research and Development (AMED) under grant
number JP19fk0108061.

**Competing interests:** The authors have declared
that no competing interests exist.

## Introduction

Carbapenems have broad spectrum activity against Gram-positive, Gram-negative, and
anaerobic bacteria and are considered agents of last-resort for complicated bacterial infec-
tions. Carbapenem-resistant Enterobacterales (CRE) are becoming increasingly prevalent
and are a global public health concern. Genes encoding carbapenemases including $bla_{KPC}$,
$bla_{IMP}$, $bla_{VIM}$, $bla_{NDM}$, $bla_{OXA-48}$, and $bla_{OXA-181}$ are encoded on plasmids, which are
transmissible between the same and different genera via conjugation [1]. A notable carba-
penemase gene is $bla_{IMP-6}$, which confers a unique susceptibility pattern to carbapenems.
Bacteria harboring $bla_{IMP-6}$ are generally resistant to the newer carbapenem, meropenem,
but are susceptible to imipenem [2,3]. The $k_{cat}/K_m$ value, as a measure of catalytic effi-
ciency of the IMP-6 protein, is approximately 7-fold higher against meropenem than
against imipenem [4] owing to the one point mutation (640-Adenine replaced by Gua-
nine), leading to a single amino acid substitution (Serine-196 replaced by Glycine) com-
pared to IMP-1 [4].

Imipenem has often been used as a representative carbapenem for antimicrobial suscepti-
bility testing. IMP-6-producing Enterobacterales isolates are occasionally overlooked in clini-
cal laboratories, and are referred to as a stealth-type [5]. Strains with $bla_{IMP-6}$ are resistant to
all other beta-lactam antimicrobials [2]. Thus, spread of these strains within and across genera
poses a serious threat to public health.

The IMP-6 carbapenemase was first isolated from *Serratia marcescens* in Japan in 1996 [4],
following the isolation of the IMP-1 enzyme from *S. marcescens* in Japan in 1991 [6]. IMP-pro-
ducing Enterobacterales have been reported mainly in Japan, Taiwan, other Asian countries,
and Australia, but have seldom been reported in European countries [7,8]. Enterobacterales
producing IMP-6 have so far been reported only from Japan [2,3,9–12]. *Pseudomonas aerugi-
nosa* producing IMP-6 have also been reported sporadically from South Korea and China
[13,14]. This is not the case for Enterobacterales. There have been sporadic reports of CRE
carrying $bla_{IMP-6}$ from hospitals located mainly in West Japan [2,3,9,10]. The most frequently
reported bacterial species harboring $bla_{IMP-6}$ are *E. coli* and *K. pneumoniae*, although the num-
ber of publications is small [15,16].

Since 2014, CRE infection has been included in the list of mandatory reporting of all symp-
tomatic cases in Japan. The National Epidemiological Surveillance of Infectious Diseases
(NESID) is a national surveillance program for symptomatic cases [17]. In 2018, 2289 CRE
cases were reported and 1684 CRE isolates were analyzed [18], which revealed that the $bla_{IMP}$
gene was the most common (254, 85.5%) out of the 297 isolates positive for carbapenemase
genes, and $bla_{IMP-6}$ and $bla_{IMP-1}$ are two dominant genes in CRE in Japan (52.0% and 46.3%
among 123 genome-sequenced isolates with $bla_{IMP}$ genes). The incidence of infections caused
by CRE with $bla_{IMP-6}$ varied by region in the report.

However, previous reports did not examine the geographical distribution of IMP-6 in each
prefecture. In addition, the NESID system involves only symptomatic cases reported by physi-
cians. Thus, the entire geographical distribution of IMP-6 including asymptomatic carriers
remains unknown.

Another national surveillance program, the Japan Nosocomial Infections Surveillance
(JANIS), has been comprehensively collecting data of all bacteria isolated from all sample types
of both symptomatic and asymptomatic patients from clinical laboratories of the participating
hospitals since 2000. The data include results of bacterial culture and antimicrobial susceptibil-
ity testing that are routinely conducted in the hospitals. The number of participating hospitals
as of January 2020 is 2223. The data covered more than 8.2 million specimens and over 5.8 mil-
lion isolates in 2018. The data stored in a national database are available for analyses in the

public interest, and will be useful to explore the entire geographical distribution of IMP-6 at phenotype level including asymptomatic carriers.

In this study, we aim to determine the geographical distribution of *E. coli* and *K. pneumoniae* isolates showing a pattern indicative of the IMP-6 phenotype (imipenem susceptible, meropenem resistant), using the JANIS database since the susceptibility pattern of $bla_{IMP-6}$-positive strains is unique. Additionally, the national data of antimicrobial usage were examined to explore a potential association between the frequency and usage of carbapenems, fluoroquinolones, and third-generation cephalosporins, which potentially causes selection of the strains with the IMP-6 phenotype.

## Materials and methods

### Data set

All inpatient data fields were extracted between January 2015 and December 2016 from the JANIS database, which stores both culture-positive and -negative test diagnostic results with all antimicrobial susceptibility testing results. Patient identifiers are de-identified by each hospital before data submission to JANIS. Approval for extraction and use of the data was granted by the Ministry of Health, Labour and Welfare (Approval no. 1010–5). We also used national data of antimicrobial usage in 2015 and 2016 tabulated for each prefecture that are publicly available at the website of AMR Clinical Reference Center in National Center for Global Health and Medicine (http://amrcrc.ncgm.go.jp/surveillance/010/20181128172333.html) (The data tabulated for each prefecture are not available after 2016). From the data, we selected total usage of carbapenems, fluoroquinolones, and third-generation cephalosporins measured as DID (Defined Daily Doses/1000 inhabitants/day).

### Data tabulation

We used a Java toolkit to extract aggregated data (stratified by specimen types) of the number of *E. coli* and *K. pneumoniae* isolates susceptible or resistant to imipenem and meropenem from the raw data in accordance with CLSI 2012 criteria [19]. De-duplication was conducted according to JANIS [20] to remove repeated isolates of the same species isolated from a patient within 30 days, regardless of specimen type, but considering the antimicrobial resistance phenotype. The de-duplication procedure selects and counts isolates with significantly different drug susceptibility as different isolates, even if they were isolated within 30 days from the same patient. A subsequent isolate is selected and counted if it shows change from susceptible to resistant (or vice versa) or a 4-fold or more comparison of the minimum inhibitory concentration value for a specific antimicrobial when compared to a previous isolate from the same patient within the 30-day period. We used an in-house Perl script to tabulate the aggregated data for each prefecture. We then calculated frequency of the IMP-6 phenotype as the number of isolates susceptible to imipenem but resistant to meropenem divided by the total number of isolates subject to antimicrobial susceptibility testing of the two antimicrobials. We also used the Java toolkit to extract aggregated data of the number of *E. coli* and *K. pneumoniae* isolates susceptible or resistant to imipenem, meropenem, ceftazidime, piperacillin, fosfomycin and amikacin. The aggregated data are available at https://github.com/bioprojects/IMP-6_phenotype_distribution.

### Statistical analyses

Descriptive statistical analyses and univariate association analyses were performed with JMP Pro version 14 (SAS Institute, Cary, NC, USA). The aggregated data of the frequency of IMP-6

phenotype in each prefecture was visualized as a map using R version 3.6.1 and leaflet package. A statistical test of association between the usage of antimicrobials (either carbapenems, fluoroquinolones, and third-generation cephalosporins) and frequency of IMP-6 phenotype was conducted by logistic mixed-effect regression using glmer function of lme4 package [21] in R. In the univariate association analyses and statistical tests, we combined the aggregated data of *E. coli* and *K. pneumoniae* to calculate frequency of IMP-6 phenotype, given that plasmids encoding *bla*$_{IMP-6}$ can transfer between genera.

## Results

Geographical distribution of frequency of *E. coli* and *K. pneumoniae* isolates with the IMP-6 phenotype is shown as prefectural maps in Fig 1. The isolates continued to be found in some prefectures, but were not in approximately half of the prefectures (white in Fig 1). The percentages of prefectures where the isolate with IMP-6 phenotype is undetectable were 42.6% and 55.3% for *E. coli* in 2015 and 2016, and 42.6% for *K. pneumoniae* in both 2015 and 2016, respectively.

More quantitatively, boxplots and histograms of the frequency distribution are shown in Fig 2. The mean of total number of antimicrobial susceptibility tests as denominator of the frequency was 4241 for *E. coli* and 2185 for *K. pneumoniae*. The distribution was obviously skewed, with a higher frequency in some prefectures. The highest frequency was 1.7% (28/1644) for *E. coli* and 1.6% (13/790) for *K. pneumoniae*.

The total number of *E. coli* isolates with the IMP-6 phenotype after de-duplication was 221 in 2015 and 242 in 2016. The corresponding numbers for *K. pneumoniae* were 189 in 2015 and 201 in 2016. A breakdown of the number of isolates according to specimen types is shown in Table 1. For *E. coli*, the isolates were most frequently from urine samples (49.3% in 2015 and 52.9% in 2016), followed by stool and respiratory samples. For *K. pneumoniae*, the isolates were almost equally frequent in urine, stool, and respiratory samples. A breakdown of the number of isolates without the de-duplication is also shown in S1 Table. No statistically significant difference in the proportion among the four specimen types was found between the two conditions (with and without de-duplication) in 2015 and 2016 for *E. coli* ($p = 0.204$ and $p = 0.795$, respectively), and for *K. pneumoniae* ($p = 0.862$ and $p = 0.478$, respectively) with chi-square test.

The relationship between the frequency and antimicrobial usage (DID) (Fig 3) clearly revealed almost no relationship between the usage of carbapenems and the frequency of the IMP-6 phenotype, while a positive relationship with third-generation cephalosporins was evident. Logistic mixed-effect regression analysis accounting for the longitudinal correlation revealed the association was statistically significant in the *E coli* and *K. pneumoniae* group ($p = 0.006$). The association was still statistically significant in *K. pneumoniae* ($p = 0.002$) and was suggested in *E. coli* ($p = 0.072$). The association between frequency of isolates with the IMP-6 phenotype and antimicrobial usage was also evident for fluoroquinolones, although the significance was weaker ($p = 0.043$ for the combined group of *E. coli* and *K. pneumoniae*) than that of third-generation cephalosporins. In the isolates with the IMP-6 phenotype, resistance rate of ceftazidime, as a representative third-generation cephalosporins, was 94.4% of the *E. coli* and 93.0% of the *K. pneumoniae* isolates across the prefectures.

## Discussion

Presently, using data of antimicrobial susceptibility testing of all isolates in more than 1,400 hospitals in Japan, isolates with the phenotype unique to IMP-6 were enumerated for each prefecture. Such a utilization of the phenotypic data of bacterial culture and antimicrobial

# *E. coli*

## 2015

## 2016

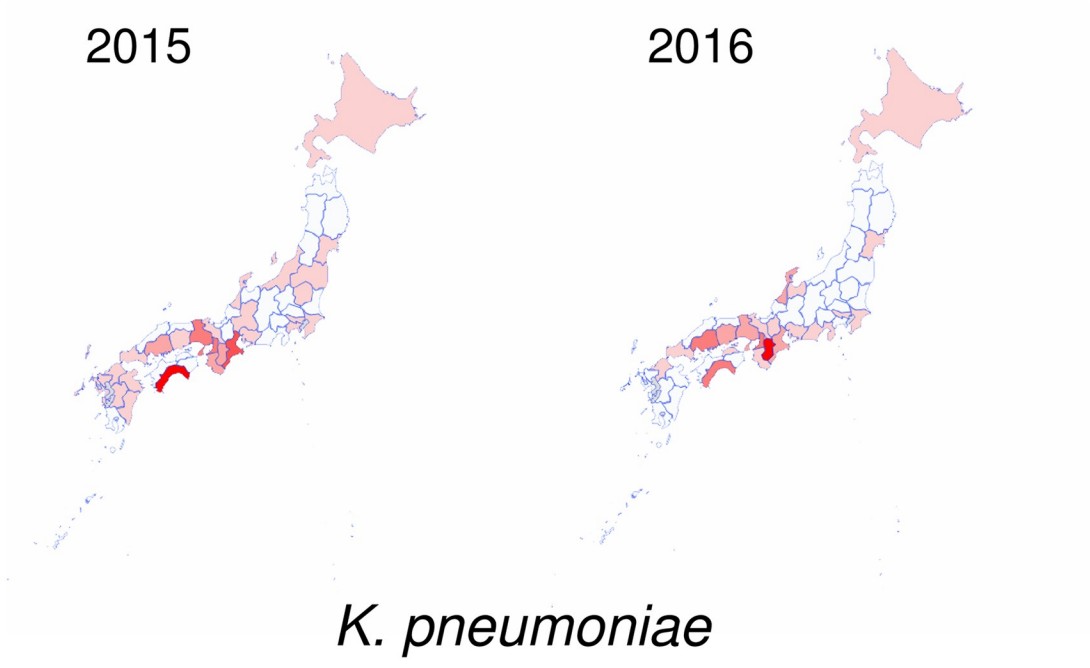

# *K. pneumoniae*

## 2015

## 2016

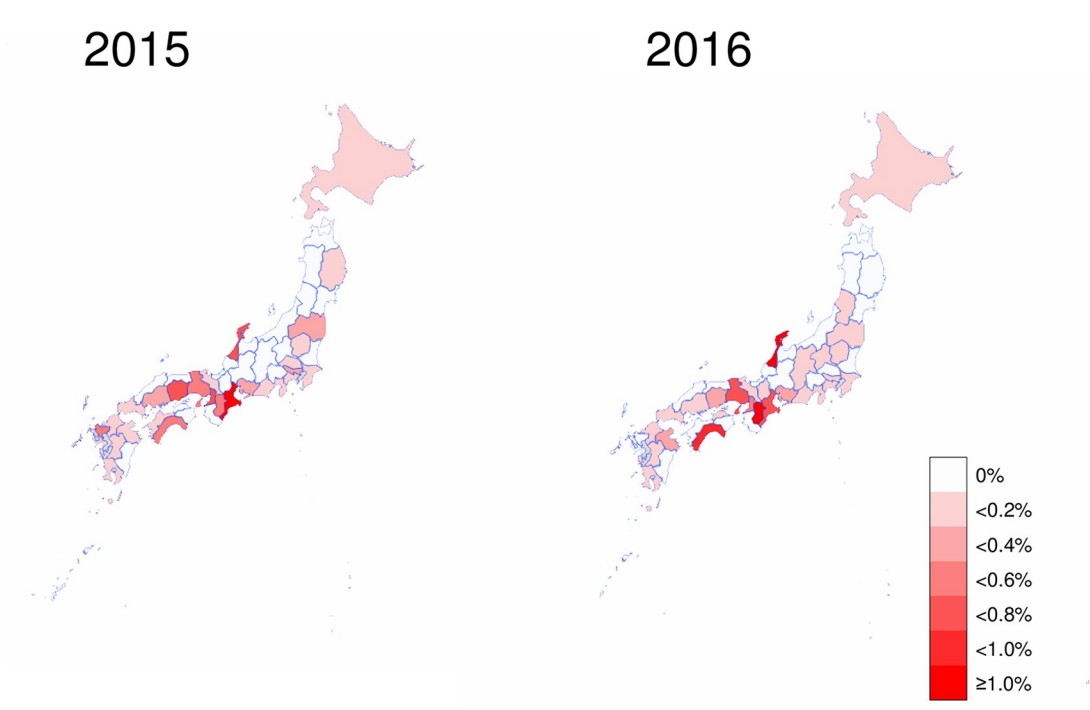

**Fig 1. Prefectural maps of frequency distribution of isolates with the IMP-6 phenotype in 2015 and 2016.** Prefectures where isolates with the IMP-6 phenotype were not isolated are colored in white, and others are colored in six different colors (with bin size 0.2%) in the red range.

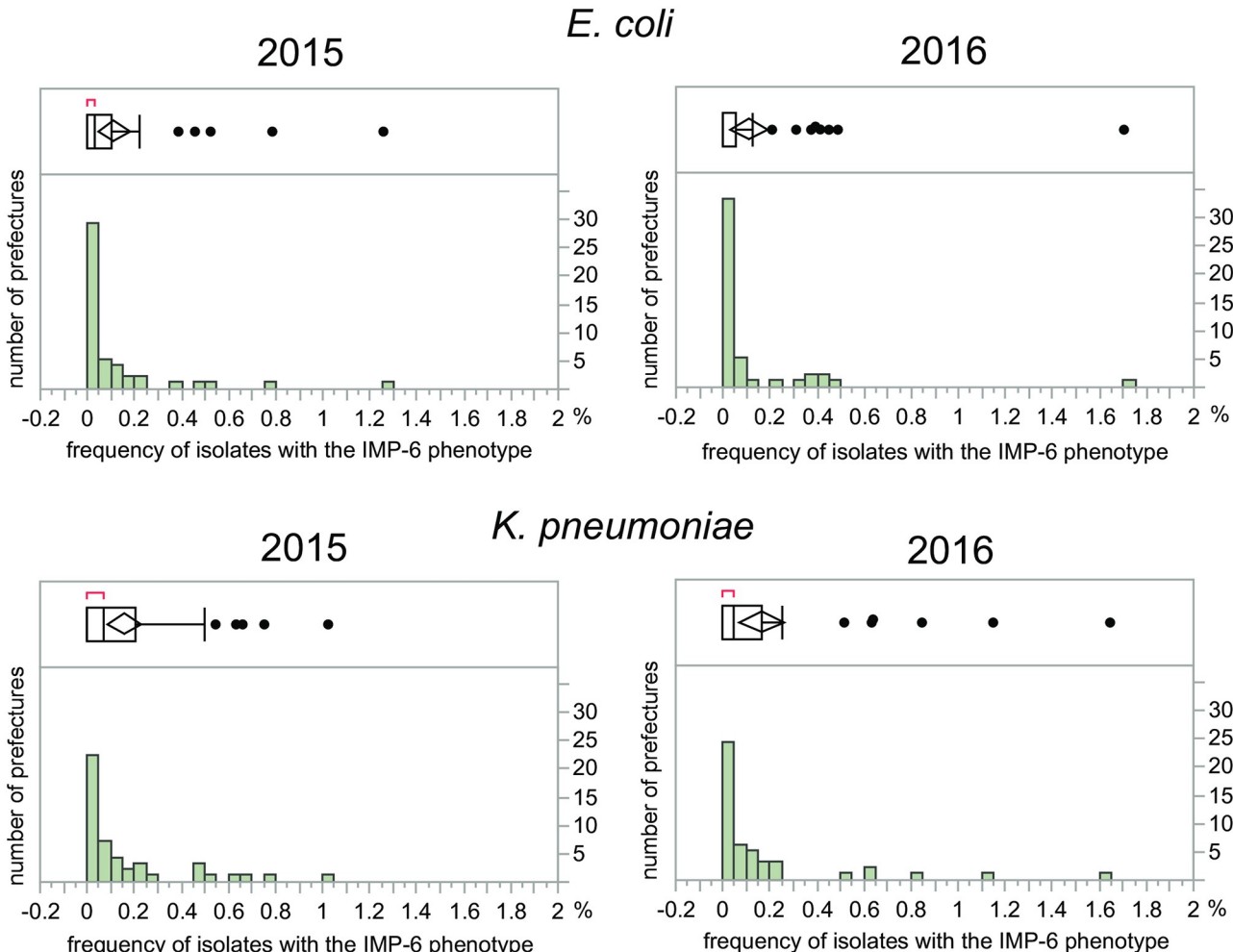

**Fig 2. Boxplots and histograms of frequency distribution of isolates with the IMP-6 phenotype in 2015 and 2016.** In the box plot, the left and right of the box indicate 25th and 75th percentile, respectively, the horizontal line indicates the median, the middle of the diamond indicates the mean, the right outliers are above the 75th percentile + 1.5 interquartile range, and the red horizontal line at the top left indicates the shortest range in which half of the data was distributed.

susceptibility testing that are routinely conducted in hospitals throughout a country is complementary and cost-effective compared to genetic testing by sequencing to detect $bla_{IMP-6}$, which requires additional time and cost and currently cannot be extended to potential asymptomatic carriers across Japan.

**Table 1. The number and proportion of isolates with the IMP-6 phenotype according to specimen types after de-duplication.**

|  | *E. coli* | | *K. pneumoniae* | |
|---|---|---|---|---|
|  | **2015** | **2016** | **2015** | **2016** |
| Blood | 7 (3.2%) | 9 (3.7%) | 17 (9.0%) | 12 (6.0%) |
| Respiratory | 46 (20.8%) | 47 (19.4%) | 72 (38.1%) | 59 (29.4%) |
| Urine | 109 (49.3%) | 128 (52.9%) | 48 (25.4%) | 70 (34.8%) |
| Stool | 59 (26.7%) | 58 (24.0%) | 52 (27.5%) | 60 (29.9%) |
| Total | 221 (100%) | 242 (100%) | 189 (100%) | 201 (100%) |

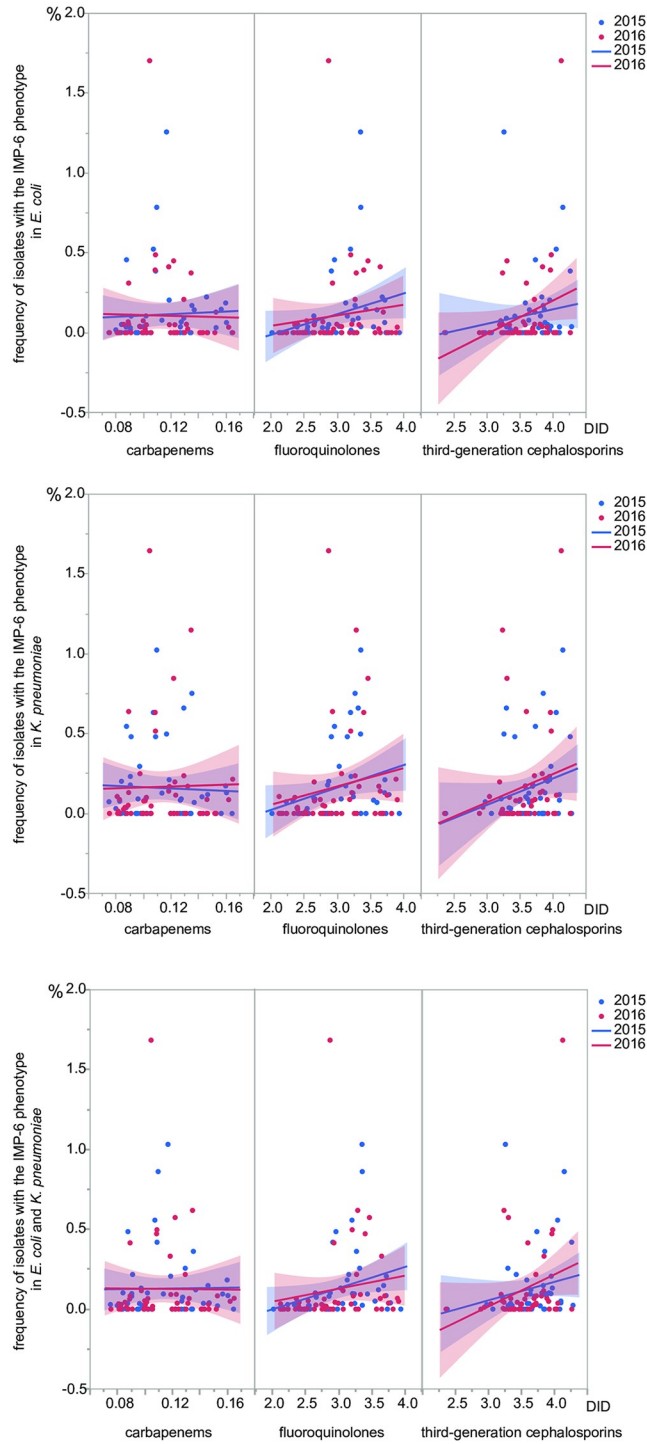

**Fig 3. Association between antimicrobial usage and frequency of isolates with the IMP-6 phenotype.** The frequency in the bottom of the figure was calculated after summing the denominator (total number of antimicrobial susceptibility testing) between *E. coli* and *K. pneumoniae* and that of the numerator (number of isolates with the IMP-6 phenotype). A linear regression line with a 95% credibility interval is illustrated for 2015 (blue) and 2016 (red).

The frequency distribution of the isolates was depicted as a map (Fig 1) and quantitatively as boxplots and histograms (Fig 2). Overall, the longitudinal correlation of the frequency between 2015 and 2016 was high, but not complete (Spearman's nonparametric correlation coefficient: 0.61 in *E. coli* and 0.72 in *K. pneumoniae*). The findings reflected that the isolates with the IMP-6 phenotype tended to be continuously found in specific prefectures, although an occasional increase in the frequency in one year was not necessarily repeated in another year. The results are consistent with the previous report of NESID that the isolates with $bla_{\text{IMP-6}}$ are detected only in some prefectures [18]. The statistically significant association suggests that third-generation cephalosporins, rather than carbapenems, might contribute to the selection of strains with the IMP-6 phenotype. The absence of an association for carbapenems might reflect that their usage is mainly confined to injections that are strictly controlled by monitoring and notification systems among the hospitals participating in the national surveillance JANIS. On the other hand, fluoroquinolones and third-generation cephalosporins are available as oral formulations that more easily administered without the involvement of the notification system, as well as injectables. As indicated by the X-axis of Fig 3, the amount of usage (measured by DID) of fluoroquinolones and third-generation cephalosporins was much larger than that of carbapenems.

It is known that a strain carrying $bla_{\text{IMP-6}}$ is resistant to third-generation cephalosporins, including ceftazidime [4]. This is confirmed by the ≥93% resistance to ceftazidime among the hundreds of isolates in our study, suggesting the phenotype corresponds well to the presence of $bla_{\text{IMP-6}}$, however, the phenotype-genotype correspondence is not absolute. The remaining 5.6% of *E. coli* and 7.0% of *K. pneumoniae* isolates examined in our study are unlikely to carry $bla_{\text{IMP-6}}$, contrary to the phenotype. Although JANIS is the comprehensive national surveillance of antimicrobial resistance, the targets thus far have been confined to bacterial culturing and antimicrobial susceptibility testing. Actual isolates and their genetic resistance determinants are not included. Further studies are warranted to elucidate the distribution of the $bla_{\text{IMP-6}}$ gene across geographical regions, and to determine the extent to which the presence of this gene corresponds to the observed antimicrobial susceptibility patterns.

A weaker association between the frequency of isolates with the IMP-6 phenotype and antimicrobial usage was evident for fluoroquinolones (Fig 3). This finding suggests that fluoroquinolone resistance might also contribute to selection of isolates with the IMP-6 phenotype. Fluoroquinolone resistance in Enterobacterales mainly involves chromosomal genes. The most commonly detected resistance mechanisms are mutations in chromosomal genes encoding gyrase and/or topoisomerase IV, such as *gyrA* and *parC*, which result in weakened quinolone-enzyme interactions and cause high-level quinolone resistance [22]. Less frequently, plasmid-encoded quinolone resistance genes, such as *qnr*, *aac(6')-Ib-cr*, and *qep*, are observed, although plasmid-mediated quinolone resistance determinants confer only low-level resistance to quinolones [2,23]. Linkage of these chromosomal mechanisms of fluoroquinolone resistance with plasmid-encoded $bla_{\text{IMP-6}}$ is expected if the plasmid harboring $bla_{\text{IMP-6}}$ is transmitted into extended-spectrum beta-lactamase (ESBL) producing *E. coli* or *K. pneumoniae* clones, such as the worldwide pandemic clone, ST131, in *E. coli* [24,25], which harbors point mutations in chromosomal *gyrA* and *parC* genes [26–28]. However, the linkage is conceivably weaker than that of $bla_{\text{IMP-6}}$ and ESBL genes encoded in the same plasmid, which could be a cause of the weaker association between frequency of isolates with the IMP-6 phenotype and usage of fluoroquinolones.

Treatment of isolates carrying $bla_{\text{IMP-6}}$ is problematic. Although isolates are susceptible to imipenem *in vitro*, the effectiveness of imipenem *in vivo* has not been established. It is known that a strain carrying $bla_{\text{IMP-6}}$ shows susceptibility to piperacillin due to the single amino acid substitution, compared to IMP-1 [4]. However, in previous reports, isolates with $bla_{\text{IMP-6}}$

almost always had ESBL genes, such as $bla_{CTX-M}$ [2,3,9,12], which renders resistance to all beta-lactams including piperacillin, except for imipenem. A report from Hiroshima, where studies on strains with $bla_{IMP-6}$ have been ongoing since 2009 [5], described sequence analyses of Enterobacterales isolates harboring $bla_{IMP-6}$. The analyses revealed that in 12/14 *K. pneumoniae*, 1/1 *K. oxytoca*, and 3/6 *E. coli*, $bla_{IMP-6}$ and $bla_{CTX-M-2}$ were encoded on the same 47-kb incN plasmid. When we checked the proportion of resistance to piperacillin in isolates with the IMP-6 phenotype, the mean results across the prefectures were 81.4% for *E. coli* and 92.6% for *K. pneumoniae*. Conversely, when we similarly checked the proportion of resistance to amikacin and fosfomycin in isolates with the IMP-6 phenotype, the mean values across the prefectures were 4.0% for *E. coli* and 4.5% for *K. pneumoniae* for amikacin, and 1.0% for *E. coli* and 4.0% for *K. pneumoniae* for fosfomycin. The resistance rates to amikacin and fosfomycin were significantly lower than the rates to piperacillin, suggesting their potential for treatment. However, characteristics, transferability, and adverse effects of these drugs need to be considered.

A systematic review and meta-analysis including 243 studies revealed the association of antimicrobial consumption with the development of antimicrobial resistance [29]. In the meta-analysis, a stronger link between consumption and resistance was found for countries in southern Europe. Analysis of some other cross-national database studies has revealed a correlation between antimicrobial consumption and resistant rate in Europe [30,31]. If possible, comparison with such international data are warranted to generalize our findings. However, antimicrobial consumption data is often complied differently between countries, and a careful data preparation is needed for accurate international comparison. Furthermore, Enterobacterales producing IMP-6 have so far been reported only from Japan. This was the reason for our focus on the inter-prefecture comparison in Japan. Further studies are also warranted to reveal appropriate measures reflecting such factors and to systematically collect and analyze them in addition to the antimicrobial usage data (S1 Text).

In summary, the frequency distribution of the isolates showing the phenotype specific to IMP-6 was determined across prefectures in Japan. The frequency of the IMP-6 phenotype was statistically significantly associated with usage of third-generation cephalosporins and was weakly associated with usage of fluoroquinolones, rather than usage of carbapenems. The results highlight the usefulness of national surveillance data of antimicrobial susceptibility testing results to infer the epidemiological distribution of isolates with specific resistance genes or mechanisms, without the additional cost and time of actual genetic testing. Such an inference is possible for other pathogens, for example Panton-Valentine Leukocidin or Toxic Shock Syndrome Toxin-1 producing methicillin-resistant *Staphylococcus aureus* [32]. The approach is also useful for exploring the emergence of a new multi-drug resistance phenotype, which requires systematic analyses of combination of resistance patterns of several key antimicrobials [33]. This study provides a basis for future studies utilizing national surveillance data of antimicrobial susceptibility testing results and integrated analyses incorporating additional data, such as antimicrobial usage and genetic testing.

## Supporting information

**S1 Table. The number and proportion of isolates with the IMP-6 phenotype according to specimen types without de-duplication.**
(DOCX)

**S1 Text.**
(DOCX)

## Acknowledgments

We are grateful to all the hospitals that participated and contributed data to JANIS, and to Editage (www.editage.jp) for English language editing.

## Author Contributions

**Conceptualization:** Koji Yahara, Motoyuki Sugai.

**Data curation:** Aki Hirabayashi, Koji Yahara.

**Formal analysis:** Aki Hirabayashi, Koji Yahara.

**Funding acquisition:** Keigo Shibayama.

**Investigation:** Aki Hirabayashi.

**Methodology:** Aki Hirabayashi, Koji Yahara.

**Project administration:** Koji Yahara, Keigo Shibayama.

**Resources:** Toshiki Kajihara, Motoyuki Sugai.

**Supervision:** Koji Yahara, Motoyuki Sugai, Keigo Shibayama.

**Validation:** Koji Yahara, Toshiki Kajihara, Motoyuki Sugai.

**Visualization:** Aki Hirabayashi, Koji Yahara.

**Writing – original draft:** Aki Hirabayashi.

**Writing – review & editing:** Aki Hirabayashi, Koji Yahara, Toshiki Kajihara, Motoyuki Sugai, Keigo Shibayama.

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
