## [Decision Letter · Decision Letter 0]

28 Aug 2020

PONE-D-20-20807

Geographical distribution of Enterobacteriaceae with a carbapenemase IMP-6 phenotype and its association with antimicrobial use: an analysis using comprehensive national surveillance data on antimicrobial resistance

PLOS ONE

Dear Dr. Hirabayashi,

Thank you for submitting your manuscript to PLOS ONE. After careful consideration, we feel that it has merit but does not fully meet PLOS ONE’s publication criteria as it currently stands. Therefore, we invite you to submit a revised version of the manuscript that addresses the points raised during the review process.

Please address all comments of the referee point by point

We look forward to receiving your revised manuscript.

Kind regards,

Iddya Karunasagar

Academic Editor

PLOS ONE

Journal Requirements:

2. Please amend the Ethics statement to include the ethics approval information mentioned in the Methods section (i.e., the name of the approval committee and the approval number). Please additionally state whether the need for informed patient consent was waived due to the retrospective nature of the study and the use of de-identified data.

Additionally, while we note that you have provided your data via a link to Github, we also ask that you provide any relevant code (i.e., the 'in-house' Perl script mentioned in the manuscript).

3. We note that [Figure(s) 1] in your submission contain [map/satellite] images which may be copyrighted. All PLOS content is published under the Creative Commons Attribution License (CC BY 4.0), which means that the manuscript, images, and Supporting Information files will be freely available online, and any third party is permitted to access, download, copy, distribute, and use these materials in any way, even commercially, with proper attribution. For these reasons, we cannot publish previously copyrighted maps or satellite images created using proprietary data, such as Google software (Google Maps, Street View, and Earth). For more information, see our copyright guidelines: http://journals.plos.org/plosone/s/licenses-and-copyright.

1.    You may seek permission from the original copyright holder of Figure(s) [#] to publish the content specifically under the CC BY 4.0 license. 

Additional Editor Comments (if provided):

There are some minor changes in the manuscript suggested by the reviewer. Please address these point by point.

Reviewers' comments:

Reviewer's Responses to Questions

**Comments to the Author**

1. Is the manuscript technically sound, and do the data support the conclusions?

Reviewer #1: Yes

2. Has the statistical analysis been performed appropriately and rigorously? 

Reviewer #1: Yes

3. Have the authors made all data underlying the findings in their manuscript fully available?

Reviewer #1: Yes

4. Is the manuscript presented in an intelligible fashion and written in standard English?

Reviewer #1: Yes

5. Review Comments to the Author

Reviewer #1: I would like to thank the editors for sending me such an interesting and relevant manuscript to review. I believe that the findings from this study are very important and have the ability not only to inform antimicrobial stewardship, but provide a good example of for other countries to investigate their own antimicrobial resistance and usage relationships.

I have a few comments for the authors, which I hope are of use.

General

- Would the authors consider using more current taxonomy i.e. Enterobacterales as opposed to Enterobacteriaceae [https://doi.org/10.1099/ijsem.0.001485]?

- Use a consistent approach to reporting p-values e.g. 1 significant figure, 2 decimal places etc.

Line 38: Suggest remove “This examination revealed a skewed frequency distribution.” This sentence means nothing in isolation.

Line 39-40: Were high rates observed in prefectures with high levels of 3GC prescribing? If so, suggest this sentence is merged with following sentence.

Line 44 and throughout: Remove “not carbapenems”. I don’t think you need to precede your 3GC observations with “not carbapenems” after the first mention.

Abstract: Could you conclude the abstract with a public health/antimicrobial stewardship recommendation? I believe the findings are important and actionable.

Line 53: Change “species” to “genera”

Line 63-65: This is a repeated statement – you have already stated that IMP carbapenemases are plasmid-mediated etc. earlier in the Introduction.

Lines 80-87: Summarise this section further. Suggest remove sentences between line 81 and 85 as you have already referenced the study and therefore detailed findings do not need to be provided.

Lines 92-96. This statement would be better placed in the Discussion, to highlight the advantages of the study design.

Lines 104-111: This reads more like an introduction of the methods. Suggest ending the Introduction by stating the aim(s) of your study.

Line 125: Defined daily dose = DDD

Line 160: Change “species” to “genera”

Lines 163-166: The detail in this sentence is superfluous since you have described the dataset and study years in the Methods. Suggest the sentence is simplified.

Line 168: Add “respectively” after “2016”.

Line 178: Provide the actual number (since this is the frequency) with the proportion in brackets immediately afterwards.

Line 179-180: This needs to precede the reporting of frequency and associated proportions. Also need to clarify what this “average” represents, and the type of “average” reported i.e. mean, median, mode?

Line 185: Change “average” to “median”.

Lines 191-192: Report 2015 data before 2016 (as you have done for the rest of the manuscript).

Lines 199-200: It is not clear which p-value relates to which specimen type.

Lines 206-210: This statement is more suited to the Discussion.

Lines 211-215: Much of this is already covered in the Methods and can be removed.

Lines 215-218: The wording of this goes beyond reporting the results, introducing elements of interpretation, which belongs in the Discussion.

Lines 241-242: If the proportion of Enterobacter cloacae is so high, why were these isolates not included in the study? An explanation in the Introduction or Methods (depending on reason) may be necessary.

Lines 242-245: This information belongs in the Results.

Lines 262-265: This information belongs in the Results.

Lines 269-271: Is there a reference for this?

Figure 1: The category “<1” does not make sense in the legend.

Figure 3: I think this data would be better presented as a table.

6. PLOS authors have the option to publish the peer review history of their article (what does this mean?). If published, this will include your full peer review and any attached files.

Reviewer #1: No

---

## [Author Response · Author response to Decision Letter 0]

23 Sep 2020

Please see the attached file "Response to Reviewers".

---

## [Editor Report · Decision Letter 1]

25 Nov 2020

Geographical distribution of Enterobacterales with a carbapenemase IMP-6 phenotype and its association with antimicrobial use: an analysis using comprehensive national surveillance data on antimicrobial resistance

PONE-D-20-20807R1

Dear Dr. Hirabayashi,

We’re pleased to inform you that your manuscript has been judged scientifically suitable for publication and will be formally accepted for publication once it meets all outstanding technical requirements.

Kind regards,

Iddya Karunasagar

Academic Editor

PLOS ONE

Additional Editor Comments (optional):

All reviewer comments addressed
---

## [Editor Report · Acceptance letter]

1 Dec 2020

PONE-D-20-20807R1 

Geographical distribution of Enterobacterales with a carbapenemase IMP-6 phenotype and its association with antimicrobial use: an analysis using comprehensive national surveillance data on antimicrobial resistance 

Dear Dr. Hirabayashi:

I'm pleased to inform you that your manuscript has been deemed suitable for publication in PLOS ONE. Congratulations! Your manuscript is now with our production department. 

Kind regards, 

on behalf of

Dr. Iddya Karunasagar 

Academic Editor

PLOS ONE